# Deep Networks with Internal Selective Attention through Feedback Connections

**Marijn F. Stollenga**[*]**, Jonathan Masci**[*]**, Faustino Gomez, Juergen Schmidhuber**
IDSIA, USI-SUPSI
Manno-Lugano, Switzerland
`{marijn,jonathan,tino,juergen}@idsia.ch`

## Abstract

Traditional convolutional neural networks (CNN) are stationary and feedforward. They neither change their parameters during evaluation nor use feedback from higher to lower layers. Real brains, however, do. So does our Deep Attention Selective Network (dasNet) architecture. DasNet's feedback structure can dynamically alter its convolutional filter sensitivities during classification. It harnesses the power of sequential processing to improve classification performance, by allowing the network to iteratively focus its internal attention on some of its convolutional filters. Feedback is trained through direct policy search in a huge million-dimensional parameter space, through scalable natural evolution strategies (SNES). On the CIFAR-10 and CIFAR-100 datasets, dasNet outperforms the previous state-of-the-art model on unaugmented datasets.

## 1   Introduction

Deep convolutional neural networks (CNNs) [1] with max-pooling layers [2] trained by backprop [3] on GPUs [4] have become the state-of-the-art in object recognition [5, 6, 7, 8], segmentation/detection [9, 10], and scene parsing [11, 12] (for an extensive review see [13]). These architectures consist of many stacked feedforward layers, mimicking the bottom-up path of the human visual cortex, where each layer learns progressively more abstract representations of the input data. Low-level stages tend to learn biologically plausible feature detectors, such as Gabor filters [14]. Detectors in higher layers learn to respond to concrete visual objects or their parts, e.g., [15]. Once trained, the CNN never changes its weights or filters during evaluation.

Evolution has discovered efficient feedforward pathways for recognizing certain objects in the blink of an eye. However, an expert ornithologist, asked to classify a bird belonging to one of two very similar species, may have to think for more than a few milliseconds before answering [16, 17], implying that several feedforward evaluations are performed, where each evaluation tries to elicit different information from the image. Since humans benefit greatly from this strategy, we hypothesise CNNs can too. This requires: (1) the formulation of a non-stationary CNN that can adapt its own behaviour post-training, and (2) a process that decides *how* to adapt the CNNs behaviour.

This paper introduces Deep Attention Selective Networks (dasNet) which model selective attention in deep CNNs by allowing each layer to influence all other layers on successive passes over an image through special connections (both bottom-up and top-down), that modulate the activity of the convolutional filters. The weights of these special connections implement a control policy that is learned through reinforcement learning *after* the CNN has been trained in the usual way via supervised learning. Given an input image, the attentional policy can enhance or suppress features over multiple passes to improve the classification of difficult cases not captured by the initially supervised

---
[*]Shared first author.

training. Our aim is to let the system check the usefulness of internal CNN filters automatically, omitting manual inspection [18].

In our current implementation, the attentional policy is evolved using Separable Natural Evolution Strategies (SNES; [19]), instead of a conventional, single agent reinforcement learning method (e.g. value iteration, temporal difference, policy gradients, etc.) due to the large number of parameters (over 1 million) required to control CNNs of the size typically used in image classification. Experiments on CIFAR-10 and CIFAR100 [20] show that on difficult classification instances, the network corrects itself by emphasising and de-emphasising certain filters, outperforming a previous state-of-the-art CNN.

## 2  Maxout Networks

In this work we use the Maxout networks [7], combined with dropout [21], as the underlying model for dasNet. Maxout networks represent the state-of-the-art for object recognition in various tasks and have only been outperformed (by a small margin) by averaging committees of several convolutional neural networks. A similar approach, which does not reduce dimensionality in favor of sparsity in the representation has also been recently presented [22]. Maxout CNNs consist of a stack of alternating convolutional and maxout layers, with a final classification layer on top:

**Convolutional Layer.**  The input to this layer can be an image or the output of a previous layer, consisting of $c$ input maps of width $m$ and height $n$: $x \in \mathbb{R}^{c \times m \times n}$. The output consists of a set of $c'$ output maps: $y \in \mathbb{R}^{c' \times m' \times n'}$. The convolutional layer is parameterised by $c \cdot c'$ filters of size $k \times k$. We denote the filters by $F_{i,j}^{\ell} \in \mathbb{R}^{k \times k}$, where $i$ and $j$ are indexes of the input and output maps and $\ell$ denotes the layer.

$$y_j^{\ell} = \sum_{i=0}^{i=c} \phi(x_i * F_{i,j}^{\ell}) \tag{1}$$

where $i$ and $j$ index the input and output map respectively, $*$ is the convolutional operator, $\phi$ is an element-wise nonlinear function, and $\ell$ is used to index the layer. The size of the output is determined by the kernel size and the stride used for the convolution (see [7]).

**Pooling Layer.**  A pooling layer is used to reduced the dimensionality of the output from a convolutional layer. The usual approach is to take the maximum value among non- or partially-overlapping patches in every map, therefore reducing dimensionality along the height and width [2]. Instead, a Maxout pooling layer reduces every $b$ consecutive maps to one map, by keeping only the maximum value for every pixel-position, where $b$ is called the block size. Thus the map reduces $c$ input maps to $c' = c/b$ output maps.

$$y_{j,x,y}^{\ell} = \max_{i=0}^{b} y_{j \cdot b + i, x, y}^{\ell-1} \tag{2}$$

where $y^{\ell} \in \mathbb{R}^{c' \times m' \times n'}$, and $\ell$ again is used to index the layer. The output of the pooling layer can either be used as input to another pair of convolutional- and pooling layers, or form input to a final classification layer.

**Classification Layer.**  Finally, a classification step is performed. First the output of the last pooling layer is flattened into one large vector $\vec{x}$, to form the input to the following equations:

$$\bar{y}_j^{\ell} = \max_{i=0..b} F_{j \cdot b + i}^{\ell} \vec{x} \tag{3}$$

$$\mathbf{v} = \sigma(F^{\ell+1} \bar{y}^{\ell}) \tag{4}$$

where $F^{\ell} \in \mathbb{R}^{N \times |\vec{x}|}$ ($N$ is chosen), and $\sigma(\cdot)$ is the softmax activation function which produces the class probabilities $\mathbf{v}$. The input is projected by $F$ and then reduced using a maxout, similar to the pooling layer (3).

# 3   Reinforcement Learning

Reinforcement learning (RL) is a general framework for learning to make sequential decisions order to maximise an external reward signal [23, 24]. The learning agent can be anything that has the ability to *act* and *perceive* in a given environment.

At time $t$, the agent receives an observation $o_t \in O$ of the current state of the environment $s_t \in S$, and selects an action, $a_t \in A$, chosen by a policy $\pi : O \to A$, where $S, O$ and $A$ the spaces of all possible states, observations, and action, respectively.[1] The agent then enters state $s_{t+1}$ and receives a reward $r_t \in \mathbb{R}$. The objective is to find the policy, $\pi$, that maximises the expected future discounted reward, $E[\sum_t \gamma^t r_t]$, where $\gamma \in [0, 1]$ discounts the future, modeling the "farsightedness" of the agent.

In dasNet, both the observation and action spaces are real valued $O = \mathbb{R}^{dim(O)}$, $A = \mathbb{R}^{dim(A)}$. Therefore, policy $\pi_\theta$ must be represented by a function approximator, e.g. a neural network, parameterised by $\theta$. Because the policies used to control the attention of the dasNet have state and actions spaces of close to a thousand dimensions, the policy parameter vector, $\theta$, will contain close to a million weights, which is impractical for standard RL methods. Therefore, we instead evolve the policy using a variant for Natural Evolution Strategies (NES; [25, 26]), called Separable NES (SNES; [19]). The NES family of black-box optimization algorithms use parameterised probability distributions over the search space, instead of an explicit population (i.e., a conventional ES [27]). Typically, the distribution is a multivariate Gaussian parameterised by mean $\mu$ and covariance matrix $\Sigma$. Each epoch a generation is sampled from the distribution, which is then updated the direction of the natural gradient of the expected fitness of the distribution. SNES differs from standard NES in that instead of maintaining the full covariance matrix of the search distribution, uses only the diagonal entries. SNES is theoretically less powerful than standard NES, but is substantially more efficient.

# 4   Deep Attention Selective Networks (dasNet)

The idea behind dasNet is to harness the power of sequential processing to improve classification performance by allowing the network to iteratively focus the attention of its filters. First, the standard Maxout net (see Section 2) is augmented to allow the filters to be weighted differently on different passes over the same image (compare to equation 1):

$$y_j^\ell = a_j^\ell \sum_{i=0}^{i=c} \phi(x_i * F_{i,j}^\ell), \tag{5}$$

where $a_j^\ell$ is the weight of the $j$-th output map in layer $\ell$, changing the strength of its activation, *before* applying the maxout pooling operator. The vector $\mathbf{a} = [a_0^0, a_1^0, \cdots, a_{c'}^0, a_0^1, \cdots, a_{c'}^1, \cdots]$ represents the action that the learned policy must select in order to sequentially focus the attention of the Maxout net on the most discriminative features in the image being processed. Changing action $\mathbf{a}$ will alter the behaviour of the CNN, resulting in different outputs, even when the image $x$ does not change. We indicate this with the following notation:

$$\mathbf{v}_t = \mathbf{M}_t(\theta, x) \tag{6}$$

where $\theta$ is the parameter vector of the policy, $\pi_\theta$, and $\mathbf{v}_t$ is the output of the network on pass $t$. Algorithm 1 describes the dasNet training algorithm. Given a Maxout net, $\mathbf{M}$, that has already been trained to classify images using training set, X, the policy, $\pi$, is evolved using SNES to focus the attention of $\mathbf{M}$. Each pass through the `while` loop represents one generation of SNES. Each generation starts by selecting a subset of $n$ images from X at random.

Then each of the $p$ samples drawn from the SNES search distribution (with mean $\mu$ and covariance $\Sigma$) representing the parameters, $\theta_i$, of a candidate policy, $\pi_{\theta_i}$, undergoes $n$ trials, one for each image in the batch. During a trial, the image is presented to the Maxout net $T$ times. In the first pass, $t = 0$, the action, $\mathbf{a}_0$, is set to $a_i = 1, \forall i$, so that the Maxout network functions as it would normally —

**Algorithm 1** TRAIN DASNET ($\mathbf{M}, \mu, \Sigma, p, n$)

```
1:  while True do
2:      images ⟸ NEXTBATCH(n)
3:      for i = 0 → p do
4:          θ_i ∼ ℕ(μ, Σ)
5:          for j = 0 → n do
6:              a_0 ⟸ 𝟙 {Initialise gates a with identity activation}
7:              for t = 0 → T do
8:                  v_t = M_t(θ_i, x_i)
9:                  o_t ⟸ h(M_t)
10:                 a_{t+1} ⟸ π_{θ_i}(o_t)
11:             end for
12:             L_i = −λ_boost d log(v_T)
13:         end for
14:         ℱ[i] ⟸ f(θ_i)
15:         Θ[i] ⟸ θ_i
16:     end for
17:     UPDATESNES(ℱ, Θ) {Details in supplementary material.}
18: end while
```

the action has no effect. Once the image is propagated through the net, an observation vector, $\mathbf{o}_0$, is constructed by concatenating the following values extracted from $\mathbf{M}$, by $h(\cdot)$:

1. the average activation of *every* output map $Avg(y_j)$ (Equation 2), of each Maxout layer.
2. the intermediate activations $\bar{y}_j$ of the classification layer.
3. the class probability vector, $\mathbf{v}_t$.

While averaging map activations provides only partial state information, these values should still be meaningful enough to allow for the selection of good actions. The candidate policy then maps the observation to an action:

$$\pi_{\theta_i}(\mathbf{o}) = dim(A)\sigma(\boldsymbol{\theta}_i\mathbf{o_t}) = \mathbf{a_t}, \tag{7}$$

where $\boldsymbol{\theta} \in \mathbb{R}^{dim(A) \times dim(O)}$ is the weight matrix of the neural network, and $\sigma$ is the softmax. Note that the softmax function is scaled by the dimensionality of the action space so that elements in the action vector average to 1 (instead of regular softmax which *sums* to 1), ensuring that all network outputs are positive, thereby keeping the filter activations stable.

On the next pass, the same image is processed again, but this time using the filter weighting, $\mathbf{a}_1$. This cycle is repeated until pass $T$ (see figure 1 for a illustration of the process), at which time the performance of the network is scored by:

$$L_i = -\lambda_{\text{boost}} d \log(\mathbf{v}_T) \tag{8}$$

$$\mathbf{v}_T = \mathbf{M}_T(\theta_i, x_i) \tag{9}$$

$$\lambda_{\text{boost}} = \begin{cases} \lambda_{\text{correct}} & \text{if } d = \|\mathbf{v}_T\|_\infty \\ \lambda_{\text{misclassified}} & \text{otherwise,} \end{cases} \tag{10}$$

where $\mathbf{v}$ is the output of $\mathbf{M}$ at the end of the pass $T$, $d$ is the correct classification, and $\lambda_{correct}$ and $\lambda_{misclassified}$ are constants. $L_i$ measures the weighted loss, where misclassified samples are weighted higher than correctly classified samples $\lambda_{misclassified} > \lambda_{correct}$. This simple form of boosting is used to focus on the 'difficult' misclassified images. Once all of the input images have been processed, the policy is assigned the fitness:

$$f(\theta_i) = \overbrace{\sum_{i=1}^{n} L_i}^{\text{cumulative score}} + \overbrace{\lambda_{L2}\|\theta_i\|_2}^{\text{regularization}} \tag{11}$$

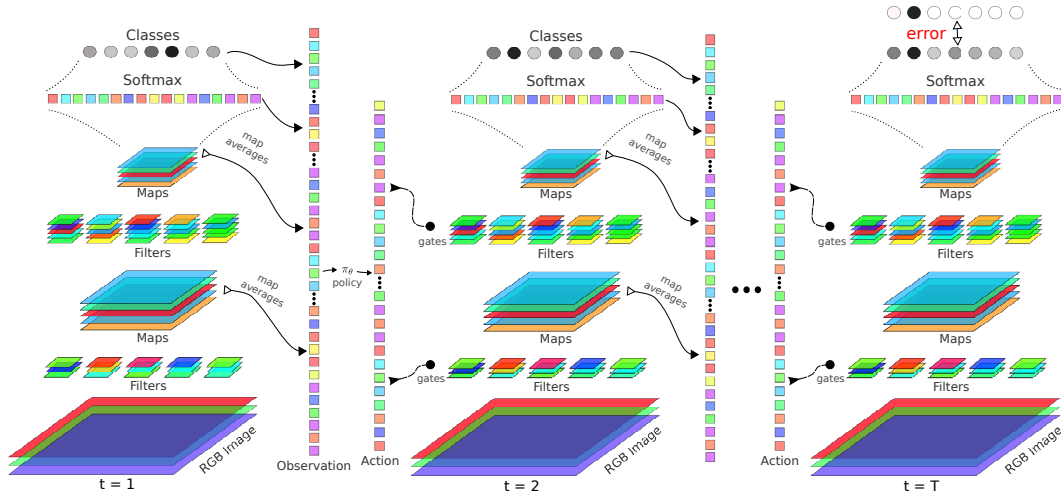

**Figure 1: The dasNet Network**. Each image in classified after $T$ passes through the network. After each forward propagation through the Maxout net, the output classification vector, the output of the second to last layer, and the averages of all feature maps, are combined into an observation vector that is used by a deterministic policy to choose an action that changes the weights of all the feature maps for the next pass of the same image. After pass $T$, the output of the Maxout net is finally used to classify the image.

where $\lambda_{L2}$ is a regularization parameter. Once all of the candidate policies have been evaluated, SNES updates its distribution parameters $(\mu, \Sigma)$ according the natural gradient calculated from the sampled fitness values, $\mathcal{F}$. As SNES repeatedly updates the distribution over the course of many generations, the expected fitness of the distribution improves, until the stopping criterion is met when no improvement is made for several consecutive epochs.

## 5 Related Work

Human vision is still the most advanced and flexible perceptual system known. Architecturally, visual cortex areas are highly connected, including direct connections over multiple levels and top-down connections. Felleman and Essen [28] constructed a (now famous) hierarchy diagram of 32 different visual cortical areas in macaque visual cortex. About 40% of all pairs of areas were considered connected, and most connected areas were connected bidirectionally. The top-down connections are more numerous than bottom-up connections, and generally more diffuse [29]. They are thought to play primarily a modulatory role, while feedforward connections serve as directed information carriers [30].

Analysis of response latencies to a newly-presented image lends credence to the theory that there are two stages of visual processing: a fast, pre-attentive phase, due to feedforward processing, followed by an attentional phase, due to the influence of recurrent processing [31]. After the feedforward pass, we can recognise and localise simple salient stimuli, which can "pop-out" [32], and response times do not increase regardless of the number of distractors. However, this effect has only been conclusively shown for basic features such as colour or orientation; for categorical stimuli or faces, whether there is a pop-out effect remains controversial [33]. Regarding the attentional phase, feedback connections are known to play important roles, such as in feature grouping [34], in differentiating a foreground from its background, (especially when the foreground is not highly salient [35]), and perceptual filling in [36]. Work by Bar et al. [37] supports the idea that top-down projections from prefrontal cortex play an important role in object recognition by quickly extracting low-level spatial frequency information to provide an initial guess about potential categories, forming a top-down expectation that biases recognition. Recurrent connections seem to rely heavily on competitive inhibition and other feedback to make object recognition more robust [38, 39].

In the context of computer vision, RL has been shown to be able to learn saccades in visual scenes to learn selective attention [40, 41], learn feedback to lower levels [42, 43], and improve face recognition [44, 45]. It has been shown to be effective for object recognition [46], and has also been

**Table 1:** Classification results on CIFAR-10 and CIFAR-100 datasets. The error on the test-set is shown for several methods. Note that the result for Dropconnect is the average of 12 models. Our method improves over the state-of-the-art reference implementation to which feedback connections are added. The recent Network in Network architecture [8] has better results when data-augmentation is applied.

| Method | CIFAR-10 | CIFAR-100 |
|---|---|---|
| Dropconnect [51] | 9.32% | - |
| Stochastic Pooling [52] | 15.13% | - |
| Multi-column CNN [5] | 11.21% | - |
| Maxout [7] | 9.38% | 38.57% |
| Maxout (trained by us) | 9.61% | 34.54% |
| **dasNet** | 9.22% | **33.78%** |
| NiN [8] | 10.41% | 35.68% |
| NiN (augmented) | **8.81%** | - |

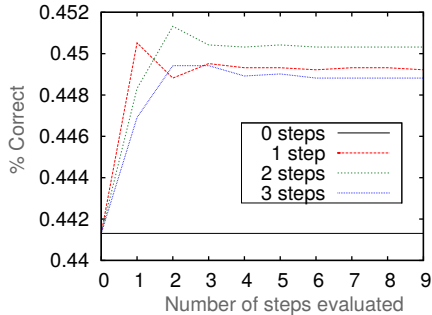

**Figure 2:** Two dasNets were trained on CIFAR-100 for different values of $T$. Then they were allowed to run for [0..9] iterations for each image. The performance peeks at the number of steps that the network is trained on, after which the performance drops, but does not explode, showing the dynamics are stable.

combined with traditional computer vision primitives [47]. Iterative processing of images using recurrency has been successfully used for image reconstruction [48], face-localization [49] and compression [50]. All these approaches show that recurrency in processing and an RL perspective can lead to novel algorithms that improve performance. However, this research is often applied to simplified datasets for demonstration purposes due to computation constraints, and are not aimed at improving the state-of-the-art. In contrast, we apply this perspective directly to the known state-of-the-art neural networks to show that this approach is now feasible and actually increases performance.

# 6 Experiments on CIFAR-10/100

The experimental evaluation of dasNet focuses on ambiguous classification cases in the CIFAR-10 and CIFAR-100 data sets where, due to a high number of common features, two classes are often mistaken for each other. These are the most interesting cases for our approach. By learning on top of an already trained model, dasNet must aim at fixing these erroneous predictions without disrupting, or forgetting, what has been learned. The CIFAR-10 dataset [20] is composed of $32 \times 32$ colour images split into $5 \times 10^4$ training and $10^4$ testing samples, where each image is assigned to one of 10 classes. The CIFAR-100 is similarly composed, but contains 100 classes. The number of steps was experimentally determined and fixed at $T = 5$; small enough to be computationally tractable while still allowing for enough interaction. In all experiments we set $\lambda_{correct} = 0.005$, $\lambda_{misclassified} = 1$ and $\lambda_{L2} = 0.005$.

The Maxout network, $\mathbf{M}$, was trained with data augmentation following global contrast normalization and ZCA normalization. The model consists of three convolutional maxout layers followed by a fully connected maxout and softmax outputs. Dropout of $0.5$ was used in all layers except the input layer, and $0.2$ for the input layer. The population size for SNES was set to $50$. Training took of dasNet took around 4 days on a GTX 560 Ti GPU, excluding the original time used to train $\mathbf{M}$.

Table 1 shows the performance of dasNet vs. other methods, where it achieves a relative improvement of $6\%$ with respect to the vanilla CNN. This establishes a new state-of-the-art result for this challenging dataset, for unaugmented data. Figure 3 shows the classification of a cat-image from the test-set. All output map activations in the final step are shown at the top. The difference in activations compared to the first step, i.e., the (de-)emphasis of each map, is shown on the bottom. On the left are the class probabilities for each time-step. At the first step, the classification is 'dog', and the cat could indeed be mistaken for a puppy. Note that in the first step, the network has not yet received any feedback. In the next step, the probability for 'cat' goes up dramatically, and subsequently drops a bit in the following steps. The network has successfully disambiguated a cat from a dog. If we investigate the filters, we see that in the lower layer emphasis changes significantly (see 'change of layer 0'). Some filters focus more on surroundings whilst others de-emphasise the eyes. In the

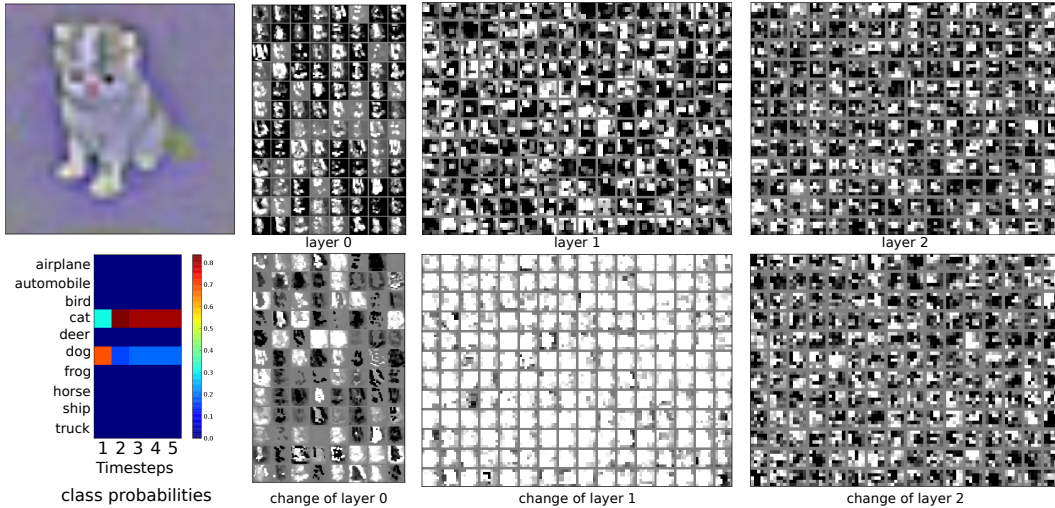

**Figure 3:** The classification of a cat by the dasNet is shown. All output map activations in the final step are shown on the top. Their changes relative to initial activations in the first step are shown at the bottom (white = emphasis, black = suppression). The changes are normalised to show the effects more clearly. The class probabilities over time are shown on the left. The network first classifies the image as a dog (wrong) but corrects itself by emphasising its convolutional filters to see it is actually a cat. Two more examples are included in the supplementary material.

second layer, almost all output maps are emphasised. In the third and highest convolutional layer, the most complex changes to the network can be seen. At this level the positional correspondence is largely lost, and the filters are known to code for 'higher level' features. It is in this layer that changes are the most influential because they are closest to the final output layers.

It is hard to qualitatively analyze the effect of the alterations. If we compare each final activation in layer 2 to its corresponding change (see Figure 3, right), we see that the activations are not simply uniformly enhanced. Instead, complex suppression and enhancement patterns are found, increasing and decreasing activation of specific pixels. Visualising what these high-level actually do is an open problem in deep learning.

**Dynamics** To investigate the dynamics, a small 2-layer dasNet network was trained for different values of $T$. Then they were evaluated by allowing them to run for $[0..9]$ steps. Figure 2 shows results of training dasNet on CIFAR-100 for $T = 1$ and $T = 2$. The performance goes up from the vanilla CNN, peaks at the $step = T$ as expected, and reduces but stays stable after that. So even though the dasNet was trained using only a small number of steps, the dynamics stay stable when these are evaluated for as many as 10 steps.

To verify whether the dasNet policy is actually making good use of its gates, we estimate their information content in the following way: The gate values in the final step are used directly for classification. The hypothesis is that if the gates are used properly, then their activation should contain information that is relevant for classification. For this purpose, a dasNet that was trained with $T = 2$. Then using *only* the final gate-values (so without e.g. the output of the classification layer), a classification using 15-nearest neighbour and logistic regression was performed. This resulted in a performance of *40.70%* and *45.74%* correct respectively, similar to the performance of dasNet, confirming that they contain significant information.

# 7   Conclusion

DasNet is a deep neural network with feedback connections that are learned by through reinforcement learning to direct selective internal attention to certain features extracted from images. After a rapid first shot image classification through a standard stack of feedforward filters, the feedback can actively alter the importance of certain filters "in hindsight", correcting the initial guess via additional internal "thoughts".

DasNet successfully learned to correct image misclassifications produced by a fully trained feedforward Maxout network. Its active, selective, internal spotlight of attention enabled state-of-the-art results. Future research will also consider more complex actions that spatially focus on (or alter) parts of observed images.

## Acknowledgments

We acknowledge Matthew Luciw for his discussions and for providing a short literature review, included in the Related Work section.

## Footnotes

[1] In this work $\pi : O \to A$ is a deterministic policy; given an observation it will always output the same action. However, $\pi$ could be extended to stochastic policies.

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
