[Supplementary Material]

# Deep Networks with Internal Selective Attention through Feedback Connections
# Supplementary Material

Marijn F. Stollenga*, Jonathan Masci*, Faustino Gomez, Juergen Schmidhuber
IDSIA, USI-SUPSI
Manno-Lugano, Switzerland
{marijn,jonathan,tino,juergen}@idsia.ch

October 31, 2014

## 1 Examples

**Figure 1:** The classification of the picture of a dog.

In the Figure 1, a very difficult classification is shown. The correct classification is dog (the dog is on the right part of the image). It can be seen that initially, without feedback, the class 'frog' has a high activation. After that, the classes 'dog', 'cat' compete and from step 3 onward, dog has a higher activation. Figure 2 shows a misclassification, where initially 'car' has a slightly higher activation, but eventually the network

**Figure 2:** The mis-classification of a car.

flips and ends up giving 'truck' a strong classification. It shows that the dynamics don't always lead to correct results.

## 2 Algorithm Details

Here we describe in more detail the UPDATESNES step. The original SNES algorithm is described in [1]. In the TRAIN DASNET function, the individuals $\theta_i$ are stored in $\Theta$ and the corresponding fitnesses $f_i$ in $\mathcal{F}$. In the actual sampling step an extra variable $s_i$ is used for each individual, from which $\theta_i$ is calculated, and used to update the parameters $\mu$ and $\Sigma$.

The sampling steps in lines 1-4 in Algorithm 1 correspond to line 4 of the algorithm in the paper. Likewise, Lines 5-9 in Algorithm 1 update the distribution parameters, and correspond to line 17 of the original algorithm.

The update parameters are set to $\eta_\mu = .8$ for the center, and $\eta_\Sigma = .3$ for the covariance. The population size is $p = 50$, and the batch size is 128.

---
**Algorithm 1** UPDATESNES

---
1: **for** $k \leftarrow 1 \cdots \cdot p$ **do**
2: $\quad s_k \sim \mathcal{N}(\vec{0}, I)$
3: $\quad \theta_k \leftarrow \mu + \Sigma s_k$
4: **end for**
5: Sort $\{(s_k, \theta_k)\}$ with respect to $f(z_k)$
6: $\nabla_\mu J \leftarrow \sum_{k=1}^p u_k \cdot s_k$
7: $\nabla_\Sigma J \leftarrow \sum_{k=1}^p u_k \cdot (s_k^2 - 1)$
8: $\mu \leftarrow \mu + \eta_\mu \cdot \Sigma \cdot \nabla_\mu J$
9: $\Sigma \leftarrow \Sigma + \exp(\eta_\sigma/2 \cdot \nabla_\Sigma J)$

---

## Footnotes

*Shared first author.

# References

[1]  T. Schaul, T. Glasmachers, and J. Schmidhuber. "High dimensions and heavy tails for natural evolution strategies". In: *GECCO*. ACM. 2011, pp. 845–852.