[Reviews · NeurIPS 2014]

Submitted by Assigned_Reviewer_24

A convnet with feedback connections to learn visual attention. The feedback is done by gating activation of some filters. The model is learned with reinforcement learning on top of a convnet. No part is particularly novel but the approach taken is a really good one in my opinion and the right way to go. And as expected they prove it with state of the art results on CIFAR. This is significant in confirming feedback/attention is the right way to go next. Quality and clarity is good.
Summary: A very good approach to vision with good results: convnets with RL-driven feedback attention. If possible, authors should go deeper on bigger problems like imagenet if they aren't already.

Submitted by Assigned_Reviewer_38

This paper adds selective attention to a convolutional neural network and develops a method for training gating parameters for attention in order to optimize the network's classification performance. It addresses an important set of problems in computer vision -- how to adaptively reconfigure complex, deep processing stages when resources are limited and different tasks need to be solved. The classification results are impressive, but the whole thing is very much a black box -- it suggests useful directions to pursue but does not reveal why the method works. The analysis of image representation is not very revealing, and there is no sense of what the learned parameters (mean and variance of the theta distribution) are doing. Nevertheless, the work is an important step beyond the current paradigm of large, feed-forward networks with parameters fixed after learning. The exposition is quite clear, though some of the equations have typos (or need to be explained better).

major comments:

I would like to see the authors seriously explore the connection to dynamical systems. The work seems closer to recurrent networks than to Reinforcement Learning. The reward is observed immediately and the policy is deterministic, so training involves a traversal through the space of recurrent "policy" parameters theta to maximize the observed objective function.

I did not find the analysis of the network processing the cat image very helpful in understanding how the new features of the model help recognition. Why are layer 1 activations all increased at the first iteration? Does this affect the objective function, or is the output invariant to global scalings? Perhaps focusing on a smaller region of the image and analyzing the evolution of the gating variables in the region could be instructive. Also, looking at difficult cases -- those misclassified by the standard Maxout network but correctly labeled by the proposed model, or teasing out the effect of the learned model parameters (mean and variance of theta) by constraining them further would be interesting.

minor comments:

Algorithm 1 description: where is h_M defined? is that the step of collecting observations? F[i] and \Theta[i] are inside the loop over images (j), are the image fitness values collected in an array or overwritten on each iteration?

Eqn 11: what is the sum over? Is Eqn 8 for a single image, and should be indexed j?

Text after eqn 11: what do 'x' and 'd' refer to?

I am not clear on why regularization of theta is necessary, since theta is already sampled from the prior and will tend to small values anyway. In fact, the last step of the optimization (gradient updates on the parameters after having sampled and evaluated the fitness of thetas) is not explained well (how do you compute the gradients) or justified theoretically (are you adjusting hyper-parameters of the posterior over policies).
Summary: Interesting approach to building deep recurrent networks for recognition. The high classification accuracy relative to standard methods suggests that recurrent computation helps extract or focus on more complex features of the images, but the black box approach does not explain why it's effective.

Submitted by Assigned_Reviewer_44

The authors proposed a CNN that incorporates a form of top-down processing in the form of attention-based feedback. The results on the CIFAR datasets show state-of-the-art performances.

This paper is generally interesting to read and demonstrates novelty in terms of model design and application for the vision problem. It is also forward looking to consider both bottom-up and top-down signals for training a CNN. I particularly like the aspect of evaluating the model across time and using gated signal to constraint the representation. Since the quality of the representations learned hinges on the attention policy, it would more interesting if the authors can suggest or demonstrate cases where the algorithm may fail.

The experiments are interesting, with the authors demonstrating the model in various ways, such as visualization to help understand it better. However, since the method is more complex than the traditional feedforward CNN, I think the paper can be improved if there were results on ILSVRC, to truly demonstrate it's scalability to a larger dataset, with larger images. That will be my main (and possibly only) concern.
Summary: The combination of attention-based feedback and and CNN makes for an interesting model, which is worth further study. More practical insights from experiments on larger data sets will be preferred.
Author Feedback
Author rebuttal: We would like to thank the reviewers for their helpful comments and good
suggestions. We try to address their responses below.

As a general comment to all reviewers, we agree that the application
to a huge dataset like ImageNet is important from the scalability
perspective, but, unfortunately, this is a significant step, that is
beyond our current hardware capabilities. Even with the latest nvidia
cards and software, ImageNet can be challenging and needs careful
design of memory management. Thus we left this for (near) future work.

Reviewer_38:
===================

> The reward is observed immediately...
We are not sure what the reviewer meant here; the reward is only observed
in the final time-step.

> I would like to see the authors seriously explore the connection to
> dynamical systems. The work seems closer to recurrent networks than
> to Reinforcement Learning.

We have discussed ourselves whether we should write from a
reinforcement learning perspective or a more dynamical/recurrent
neural network perspective. We chose the first, mainly because if we
view it as an RNN, we would apply 'backpropagation through time', but
this is infeasible due to the massive memory usage that would require.
From the RL perspective direct policy search is a logical
choice and is actually feasible as we showed.

We also considered which
direction we want this research to go, and future work will involve
more elaborate 'actions' (i.e. spatial focus, 'where to look', or even
involve external world). Then you can't (easily) work with gradients
through time and the RNN perspective falls a bit flat. We wanted to
reflect this in the current paper, and make the flow for future papers
more logical.

> I did not find the analysis of the network processing the cat image
> very helpful in understanding how the new features of the model help
> recognition.

We agree that it is hard to see exactly `what' the policy
has learned, from the figure. Much more work needs to been done,
including that which is suggested by the reviewer to investigate the
internal dynamics of the network decision making process.

> I am not clear on why regularization of theta is necessary, since
> theta is already sampled from the prior and will tend to small values
> anyway. In fact, the last step of the optimization (gradient updates
> on the parameters after having sampled and evaluated the fitness of
> thetas) is not explained well (how do you compute the gradients) or
> justified theoretically (are you adjusting hyper­parameters of the
> posterior over policies).

Theta is sampled from a distribution, but these distribution
parameters change each generation of the NES algorithm, thus
theta is not regularized implicitly. We determined empirically
that explicit regularization helps to speed up search and prevent overfitting.
We did not go into the details of the natural gradient
computation of NES algorithm as it is somewhat involved
and off-topic, but it is very principled and is described fully in
several of the cited papers.